# Regional impacts of black carbon morphologies on shortwave aerosol-radiation interactions: A comparative study between the US and China

Jie Luo[1,2], Zhengqiang Li[1], Chenchong Zhang[3], Qixing Zhang[2,*], Yongming Zhang[2], Ying Zhang[1], Gabriele Curci[5,6], and Rajan K. Chakrabarty[3,4]

[1]State Environment Protection Key Laboratory of Satellite Remote Sensing, Aerospace Information Research Institute, Chinese Academy of Sciences, Beijing 100101, China
[2]State Key Laboratory of Fire Science, University of Science and Technology of China, Hefei, Anhui 230026, China
[3]Center for Aerosol Science and Engineering, Department of Energy, Environmental and Chemical Engineering, Washington University in St. Louis, St. Louis, MO 63130, USA
[4]McDonnell Center for the Space Sciences, Washington University in St. Louis, St. Louis, MO 63130, USA
[5]Department of Physical and Chemical Sciences, University of L'Aquila, L'Aquila, Italy
[6]Center of Excellence in Telesening of Environment and Model Prediction of Severe Events (CETEMPS), University of L'Aquila, L'Aquila (AQ), Italy

**Correspondence:** Qixing Zhang (qixing@ustc.edu.cn)

**Abstract.** Black carbon (BC) is one of the dominant absorbing aerosol species in the atmosphere. It normally has complex fractal-like structures due to the aggregation process during combustion. A wide range of aerosol-radiation interactions (ARI) of BC has been reported throughout experimental and modeling studies. One reason for the large discrepancies among multiple studies is the application of the over-simplified spherical morphology for BC in ARI estimates. In current climate models, the Mie theory is commonly used to calculate the optical properties of spherical BC aerosols. Here, we employ a regional chemical transport model coupled with a radiative transfer code that utilizes the non-spherical BC optical simulations to re-evaluate the effects of particles' morphologies on BC shortwave ARI, and the wavelength range of $0.3 – 4.0 \ \mu m$ was considered. Anthropogenic activities and wildfires are two major sources of BC emissions. Therefore, we choose the typical polluted area in eastern China which is dominated by anthropogenic emissions, and the fire region in the northwest US which is dominated by fire emissions in this study. A one-month simulation in eastern China and a seven-days simulation in the fire region in the northwest US were performed. The fractal BC model generally presents a larger clear-sky ARI compared to the spherical BC model. Assuming BC particles are externally mixed with other aerosols, the relative differences in the time-averaged clear-sky ARI between the fractal model with a fractal dimension ($D_f$) of 1.8 and the spherical model are 12.1% – 20.6% and 10.5% – 14.9% for typical polluted urban cities in China and fire sites in the northwest US, respectively. Furthermore, the regional-mean clear-sky ARI is also significantly affected by the BC morphology, and relative differences of 17.1% and 38.7% between the fractal model with a $D_f$ of 1.8 and the spherical model were observed in eastern China and the northwest US, respectively. However, the existence of clouds would weaken the BC morphological effects. The time-averaged all-sky ARI relative differences between the fractal model with a $D_f$ of 1.8 and the spherical model are 4.9% – 6.4% and 9.0% – 11.3% in typical urban polluted cities and typical fire sites, respectively. Besides, for the regional-mean all-sky ARI, the relative

differences between the fractal model and the spherical model are less than 7.3% and 16.8% in the polluted urban area in China and the fire region in the US, respectively. The results imply that current climate modeling may significantly underestimate the BC ARI uncertainties as the morphological effects on BC ARI are ignored in most climate models.

## 1 Introduction

Black carbon (BC), as the main absorbing aerosol in the atmosphere, exerts a positive radiative forcing, and lofts smoke plumes (Buseck and Buseck, 2000; Streets et al., 2006; Moosmüller et al., 2009). However, there are still large uncertainties in evaluating the BC radiative effects. An important cause of the discrepancy is BC's complex morphology. BC aerosols are assumed as spheres, and the optical properties are calculated using the Mie theory in most climate and atmospheric chemical transport models, such as Community Earth System Model (CESM) (Danabasoglu et al., 2020), the Model for Interdisciplinary Research on Climate (MIROC-SPRINTARS) (Takemura et al., 2005, 2009), Weather Research and Forecasting coupled to Chemistry (WRF-Chem) (Grell et al., 2005; Fast et al., 2006), and GEOS-Chem. However, many studies have shown that BC particles, especially those nascent ones, have fractal-like structures. The spherical assumption for BC can lead to a large deviation from the field measurement data and non-spherical simulated results (Chakrabarty et al., 2007; Luo et al., 2018c; He et al., 2015; Liu and Mishchenko, 2005; Luo et al., 2018d; Mishchenko et al., 2016a; Luo et al., 2021b). Based on the sampled BC images, researchers found that the shape of uncoated BC aggregates can be fitted well by a fractal law with monomer number ($N_s$), mean monomer radius ($R$), fractal prefactor ($k_0$), fractal dimension ($D_f$) and the radius of gyration ($R_g$) (Mishchenko et al., 2002; Sorensen, 2011; Luo et al., 2021a):

$$N_s = k_0 (\frac{R_g}{R})^{D_f} \tag{1}$$

Previous studies have shown that aggregated particle models are more realistic to reproduce the optical measurement results (Kahnert, 2010a; Luo et al., 2019, 2018b). Some studies have used the fractal BC models to investigate the radiative properties of BC (Wu et al., 2015; Liu et al., 2015a; Liu and Mishchenko, 2005; Yin and Liu, 2010; Teng et al., 2019; Luo et al., 2018a; Kahnert, 2010a). The direct radiative effect (DRE) was widely used to evaluate the climate effects of aerosol (Bond et al., 2013; Saleh et al., 2015). IPCC (2014) suggested using the new terminology of the aerosol-radiation interactions (ARI) instead of DRE. Thus, in this work, we use the terminology of ARI to replace DRE. However, an extremely limited number of studies have evaluated the ARI of non-spherical BC in regional or global climate models. Kahnert (2010b) has made efforts to simulate the radiative properties of freshly emitted BC using the Multiple-scale Atmospheric Transport and CHemistry (MATCH) model. That study assumed a fixed solar zenith angle (SZA) and restricted the modeling region in Western Europe. Expanding the modeling range to regions with different emission characteristics is important to understand the effects of BC sources on ARI.

A global mean BC all-sky ARI of +0.6 W m$^{-2}$ has been reported by IPCC (2014). However, the BC ARI values estimated based on in-situ optical measurements in some regions are much larger than the rest. BC emissions in China roughly account for one-fourth of its global anthropogenic emission budget (Streets et al., 2001). Eastern China, a typical polluted region, is dominated by anthropogenic emissions (Zhang et al., 2009; Li et al., 2017a). Therefore, BC ARI in eastern China has gained increasing interest.

Besides the anthropogenic sources, wildfires significantly contribute to the regional BC emissions. The extremely-high BC concentrations can be found in those fire sites. Recent studies have shown that BC emitted from fire sites can also loft the surrounding atmospheric aerosols to the upper troposphere and lower stratosphere (Yu et al., 2019). Thus, the investigation of

BC ARI in these regions is important to understand the detailed plume dynamics and the warming effects of BC. Northwest US, as one of the regions where the wildfire is most frequent, has also been investigated in addition to eastern China.

In this work, we employed WRF-Chem to simulate the aerosol mass concentrations. Note here that WRF-Chem assumes aerosols to be spherical. Therefore, the radiative parameters of fractal BC aggregates were calculated offline using an optical module, Flexible Aerosol Optical Depth (FlexAOD) (Curci, 2012). We calculated the ARI at the top of the atmosphere (TOA)

using a radiative transfer model, libRadtran (Mayer and Kylling, 2005), after the particles' optical properties were calculated.

## 2    Method

### 2.1    Aerosol distribution simulation

In this work, WRF-Chem version 4.1.3 was used to simulate the transport of atmospheric species. Two areas were selected to represent the BC sources with different emission characteristics. Eastern China, a major polluted region in the world, represents

the typical polluted urban region. It consists of 115 east-west grids and 105 south-north grids centered at 112.00°E, 35.00°N with a grid resolution of 18 km. The northwest US, one of the regions where the wildfire is most frequent in the world, was also investigated in this work. The fire region consists of 120 east-west grids and 120 south-north grids centered at 121.48°W, 39.89°N with a grid resolution of 4 km. The schematics of the two regions are shown in Figure S1. Both regions have 33 vertical layers above the ground, with a top pressure of 50 hPa.

We used the Model of Emissions of Gases and Aerosols from Nature version 2.1 (MEGAN2.1) to compute the biogenic emissions over two regions (Guenther et al., 1994, 2006). The anthropogenic inventory for eastern China in the year 2016 was compiled by the Multi-resolution Emission Inventory for China (MEIC) (Li et al., 2014; Liu et al., 2015b). We used the MIX anthropogenic inventory for the region outside China (Li et al., 2017b). The Regional Acid Deposition Model version 2 (RADM2) atmospheric chemical mechanism (Stockwell et al.) and the Model Aerosol Dynamics for Europe with the Sec-

ondary ORGanic Aerosol Model (MADE/SORGAM) were applied in the simulation of eastern China (Seinfeld et al., 2001; Ackermann et al., 1998). Fast-J photolysis scheme (Wild et al., 2000) was used to simulate the photolysis rates. The physical scheme options in WRF-Chem are shown in Table S1. The simulations in eastern China started at 0:00 UTC on November 31st, 2016, and ended at 00:00 UTC on January 1st, 2017. The data from 0:00 UTC on December 1st, 2016 to 00:00 UTC on January 1st, 2017 was used for analysis.

The Emission Database for Global Atmospheric Research for Hemispheric Transport of Air Pollution (EDGAR-HTAP) version 2 emission inventory for 2010 was used in the northwest US. MOSAIC aerosol scheme (Zaveri and Peters, 1999; Zaveri et al., 2008) and the Carbon Bond Mechanism Z (CBM-Z) photochemical mechanism (Zaveri and Peters, 1999) were used in the fire region simulation. The Fire emission was provided by the Fire INventory from NCAR (FINN) (Wiedinmyer et al., 2011). Note here that EDGAR-HTAP anthropogenic inventory and FINN were provided for the MOZART chemical mecha-

nism, so we mapped the emission for the Model for Ozone and Related chemical Tracers (MOZART) chemical mechanism to the CBM-Z chemical mechanism based on the study of Emmons et al. (2010). For both simulations in eastern China and the northwest US, the National Center for Environmental Prediction (NCEP) Global Forecast System (GFS) Final Analysis (FNL) with a horizontal grid spacing of 0.25° and 6 h intervals was used to provide the meteorological initial and boundary conditions. The chemical initial and boundary conditions were obtained from the Model for Ozone and Related chemical Tracers, version 4 (MOZART-4). The simulations in the northwest US started at 0:00 UTC on August 5th, 2016, and ended at 00:00 UTC on August 21st, and the data from 0:00 UTC on August 14th to 00:00 UTC on August 21st was used for analysis.

## 2.2 The morphology of BC

In this work, we only consider externally mixed BC aerosols, which are commonly represented by fractal structures. $D_f$ is a key parameter to describe the compactness of fractal BC (Wang et al., 2017; Yuan et al., 2019). $D_f$ increases from approximately 1.8 to 3 when the BC morphology can vary from a chain-like structure to a spherical structure. The freshly emitted BC generally exhibits a fluffy structure with a $D_f$ of approximately 1.8 (Heinson et al., 2010, 2017). The laboratory measurements also showed that the freshly emitted BC generally presents a small $D_f$. Chakrabarty et al. (2006) have shown that $D_f$ of BC emitted from wildland fuels generally exhibits a range of 1.67 – 1.83. A $D_f$ range of 1.6 – 1.9 was observed for BC produced from diesel combustion (Wentzel et al., 2003). China et al. (2013b) indicated that the BC freshly emitted from wildfire generally exhibits $D_f$ with a range of 1.74 – 1.92. However, a more compact structure was commonly observed for BC in the atmosphere with the particle aging (Li et al., 2003; Adachi et al., 2014; Chen et al., 2016; Adachi et al., 2010). A $D_f$ range of 2.2 – 2.4 was observed in the study of Adachi et al. (2010). The fractal structures with a larger $D_f$ are widely used to describe the BC with more compact structures (Adachi et al., 2010). Chakrabarty et al. (2006) further showed that the $D_f$ of aged BC can reach up to 2.6. To represent both fluffy and compact BC, $D_f$s of 1.8, 2.2, and 2.6 were considered in this study. While the fractal prefactor $k_0$ was also measured in a wide range in the atmosphere, its impact on the optical properties was relatively small. We assumed a fixed $k_0$ of 1.2 in this work. The typical morphologies of fractal BC are shown in Figure 1.

The volume-mean particle radius was commonly used to describe the size of non-spherical BC. Previous studies have observed a range of approximately 8 – 57 nm for BC monomer radius (Eggersdorfer and Pratsinis, 2012; Mikhailov et al., 2006; KÖylü and Faeth, 1992; Lee et al., 2002), while Kahnert and Kanngießer (2020) further showed that the typical range is approximately 10 – 25 nm. In this work, we assumed a fixed monomer radius of 20 nm. We considered an $N_s$ range of 1 – 1000 to represent BC with different sizes. The volume-mean particle radius ($r_p$) can be calculated using:

$$r_p = RN_s^{1/3} \tag{2}$$

Note here that BC can also internally mix with other compositions, and the morphology can become more complex (Wang et al., 2021c, 2017). However, we mainly focus on the freshly emitted BC, and only consider externally mixed BC. Further investigations would be performed for more complex internally mixed BC in the future.

## 2.3 The refractive index and size distribution of BC

BC refractive index shows spectral dependence (Chang and Charalampopoulos, 1990), while it doesn't vary largely with wavelengths in the short wavelength range (Liu et al., 2018; Lack and Cappa, 2010; Bond and Bergstrom, 2006). The suggested BC refractive index values by Bond and Bergstrom (2006) were commonly used. In this work, the median value of $1.85 + 0.71i$ was used, as it was widely used in many regional and global climate models (e.g. WRF-Chem). The size distribution of BC also suffers large uncertainties from different fuels and conditions. The size distribution of BC is commonly fitted by a lognormal size distribution with a geometric mean radius ($r_g$), and a geometric standard deviation ($\sigma_g$) (Schwarz et al., 2008; Mishchenko et al., 2016b):

$$n(r_p) = \frac{N_0}{\sqrt{2\pi}r_p ln(\sigma_g)} exp\left[-\left(\frac{ln(r_p) - ln(r_g)}{\sqrt{2}ln(\sigma_g)}\right)^2\right] \tag{3}$$

where $n(r_p)$ is the probability density distribution of particle numer concentrations; $r_p$ is the volume-mean particle radius, $N_0$ is the number concentration, which can be calculated by the mass concentration obtained from WRF-Chem by assuming BC mass density, $r_g$, and $\sigma_g$. The details about the calculation of $N_0$ are shown in Curci (2012).

BC geometric mean radius of $0.05 - 0.06~\mu m$ is frequently observed by instruments and widely assumed in numerical studies (Alexander et al., 2008; Coz and Leck, 2011; Reddington et al., 2013; Liu et al., 2018). In this work, BC geometric mean radius was assumed to be $0.05~\mu m$, and $\sigma_g$ was assumed to be 1.6. We used the volume-equivalent radius of to characterize the particle size of factal BC. The density of BC was assumed to be $1.8~gm^{-3}$ based on the suggested values by Bond and Bergstrom (2006).

## 2.4 BC radiative properties

In this work, BC radiative properties were calculated using the Multiple Sphere T-matrix Method (MSTM) (Mackowski and Mishchenko, 2011, 1996). The MSTM can efficiently calculate the optical properties of spheres without intersecting surfaces. The MSTM has high computational efficiency because it theoretically calculates the optical properties of randomly oriented particles without numerically averaging them over different particle orientations. The MSTM can directly calculate the extinction efficiency ($Q_{ext}$), scattering efficiency ($Q_{sca}$), and phase function ($P$) with the refractive index, wavelength, and the input shapefile. Then, the extinction cross-section ($C_{ext}$) and scattering cross-section ($C_{sca}$) were obtained using:

$$C_{ext} = Q_{ext}\pi r_p^2 \tag{4}$$

$$C_{sca} = Q_{sca}\pi r_p^2 \tag{5}$$

Extinction coefficient ($b_{ext}$), Scattering coefficient ($b_{sca}$) and bulk phase function <P> were calculated using the following equations:

$$b_{ext} = \int_{r_{min}}^{r_{max}} C_{ext}(r_p)n(r_p)dr \tag{6}$$

$$b_{sca} = \int_{r_{min}}^{r_{max}} C_{sca}(r_p)n(r_p)dr \tag{7}$$

$$< P(\theta) >= \frac{\int_{r_{min}}^{r_{max}} C_{sca}(r_p)P(\theta,r_p)n(r_p)dr}{b_{sca}} \tag{8}$$

In climate modeling, instead of using the phase function, the Legendre expansion coefficients were commonly used:

$$< P(\theta) >= \sum_{S=1}^{S_{max}} \alpha_S P_S(cos\theta) \tag{9}$$

where $\theta$ is the scattering angle, $P_S$ are generalized spherical functions, $\alpha_S$ are the Legendre expansion coefficients, and $S_{mas}$ is the order of truncation of the Legendre expansion coefficients.

In this work, we used the pmom tool which is available in libRadtran software for calculating the Legendre expansion coefficients. With the inputs of the aerosol bulk phase function and the desired number of Legendre expansion coefficients, the pmom tool can calculate the Legendre expansion coefficients.

The radiative properties of fractal BC and spherical BC were calculated at 300 nm – 4000 nm wavelengths. The step size of $\Delta\lambda$ = 50 nm was chosen when $\lambda$ is less than 1000 nm, while $\Delta\lambda$= 200 nm was selected for 1000 nm $\leq \lambda \leq$ 2000 nm, and $\Delta\lambda$= 400 nm when $\lambda \geq$ 2000 nm. We created look-up tables for $b_{ext}$, $b_{sca}$, and the Legendre expansion coefficients of phase functions for each $\sigma_g$ and $r_g$. Thus, the optical properties of BC can be obtained by interpolating the look-up tables.

## 2.5  Flexible Aerosol Optical Depth (FlexAOD)

The aerosol mass concentrations from WRF-Chem were inputted into an optical software Flexible Aerosol Optical Depth (FlexAOD) (Curci, 2012; Curci et al., 2015) to calculate the aerosol radiative properties. FlexAOD is an optical post-processing tool for the atmospheric chemistry-transport model, and it started as a tool for the GEOS-Chem model. We have made some modifications to FlexAOD to make it accommodate the WRF-Chem outputs. We first mapped the aerosols from WRF-Chem into 5 categories: BC, organic carbon(OC), inorganic salt (INS), sea salt (SA), and dust (DST). The mapping details are shown in Tables S2 – S3. After the WRF-Chem species were mapped, the size distribution, refractive indices, and hygroscopic growth factors were then assigned.

FlexAOD firstly reads the aerosol mass concentrations from WRF-Chem, then converts them to aerosol volume concentrations based on the assigned mass densities. Based on the assigned normalized size distributions, we can calculate the number concentration ($N_0$) of each aerosol. FlexAOD pre-calculates the optical properties of each type of aerosol by assuming $N_0$ = 1 with the assumed size distributions. The total scattering/extinction coefficients can be obtained by multiplying the pre-calculated scattering/extinction cross-sections with the number concentrations. The phase function of each type of aerosol is identical to the pre-calculated phase function by assuming $N_0$ = 1, then the total phase function was calculated according to the number concentration of each aerosol. In FlexAOD, aerosol shapes were assumed to be spherical and the corresponding

optical properties of each aerosol species were calculated using the Mie code provided by Mishchenko et al. (1999). The bulk optical properties were then calculated by combining an assembly of aerosols.

BC optical properties were overwritten using the look-up tables created in Section 2.4. As described in Section 2.4, we have created look-up tables for non-spherical BC with different size distributions. Thus, if the size distribution of BC is assigned, the optical properties of BC with a normalized size distribution can be determined by interpolating the look-up tables. Once the number concentration is calculated, we can determine the total optical properties.

Apart from BC, the physical properties of other chemical species were also specified. We used the OC refractive indices suggested by Highwood (2009). The density of OC varies under different conditions. The density of the oxidized organic aerosol was reported to be approximately 1.3 g cm$^{-3}$ (Cross et al., 2007), while Nakao et al. (2013) reported that the density of OC with lower oxidation was approximately $1 - 1.2$ g cm$^{-3}$. For freshly formed OC, $0.9 - 1.1$ g cm$^{-3}$ was used by Liu et al. (2017b). In this work, the density of OC was assumed to be 1.2 g m$^{-3}$. OC size is also commonly fitted by a lognormal size distribution. In the study of He et al. (2016) and Dentener et al. (2006), $r_g = 0.03$ $\mu$m and $r_g = 0.075$ $\mu$m were assumed for hydrophobic and hydrophilic OC, respectively. In this study, all OC was assumed to be hydrophilic, and we assumed a $r_g$ of 0.075 $\mu$m and a $\sigma_g$ of 1.6 for OC. The refractive indices of dust were identical to those used in The Goddard Chemistry Aerosol Radiation and Transport (GOCART) Model (Chin et al., 2002). For dust, the gamma distribution was assumed: (Martin et al., 2003; Curci, 2012):

$$n(r_p) = N_0 r_p^{(1-3b)/b} exp\left(-\frac{r_p}{ab}\right) \tag{10}$$

where $a$ and $b$ are two parameters for the distribution, and $b$ is in the range of $0 - 0.5$.

The refractive indices of other chemical species were adapted from the Optical Properties of Aerosols and Clouds (OPAC) package (Hess et al., 1998). The physical properties are displayed in Table S4. Similar to the study of Curci et al. (2019), the hygroscopic growth factors of different aerosols were taken from the OPAC package (Hess et al., 1998). Note here that many internally mixed particles exist in the atmosphere, while in this study we mainly aim to study the morphological effects of freshly emitted particles, and more complex particles may be investigated in the future. Effective refractive indices were calculated using the volume mixing method for hydrophilic particles.

The total column single-scattering albedo (SSA) and Aerosol Optical Depth (AOD) were calculated by FlexAOD, and Absorption Aerosol Optical Depth (AAOD) was calculated by:

$$AAOD = AOD(1 - SSA) \tag{11}$$

## 2.6 ARI modeling

The optical properties (extinction coefficient, SSA, asymmetric factor (ASY)) calculated using FlexAOD at each WRF-Chem grid were inputted into a radiative transfer model, libRadtran (Mayer and Kylling, 2005), to calculate the radiative fluxes at the top-of-the-atmosphere (TOA). The radiative transfer equation was solved by DIScrete Ordinate Radiative Transfer (DISORT) radiative transfer equation solver (Stamnes et al., 1988; Buras et al., 2011). The libRadtran can select a standard atmosphere background and determine the solar zenith angle (SZA) based on the longitude, latitude, and UTC time. The surface albedo

information was obtained from NASA Earth Observations (NEO). The radiative transfer calculations were performed for each hour and then were averaged over one day. In this work, ARI of BC aerosol were calculated using following equations:

$$\mathrm{ARI} = \mathrm{FLUX_{With\ BC}} - \mathrm{FLUX_{Without\ BC}} \tag{12}$$

$$\mathrm{FLUX} = \mathrm{F}^{\downarrow} - \mathrm{F}^{\uparrow} \tag{13}$$

where $\mathrm{F}^{\downarrow}$ represents downward radiative flux and $\mathrm{F}^{\uparrow}$ represents upward radiative flux. In this work, we just considered the ARI at the TOA. Only shortwave ARI was considered, and the wavelength is in the range of $0.3 - 4.0\ \mu m$.

In this work, both the clear-sky ARI and all-sky ARI were calculated. The daily-mean cloud optical thickness, cloud effective radius, and cloud cover data from the MOderate Resolution Imaging Spectroradiometer (MODIS) products were used for the all-sky ARI calculations. The regional-mean BC ARI was also calculated. The aerosol optical properties and cloud properties were firstly averaged, then the radiative transfer calculations were performed for only one day (the median day), and similar methods were also used in previous studies (eg. Saleh et al. (2015); Tuccella et al. (2020a)).

## 3 Results

### 3.1 Impacts of BC morphology on AOD and AAOD

To verify the modeling performance of the aerosol concentrations, we compared the simulated $PM_{2.5}$ concentrations with observations at some monitoring sites, and the results are shown in Figure 2. In the figures, the left column represents the typical cities in eastern China, and the right column represents the sites in the northwest US. The calculated $PM_{2.5}$ concentrations in eastern China are generally consistent with the observations. Even though the simulated $PM_{2.5}$ concentrations in the fire region are a little higher than the observations, the deviations are not large, and the general trends are consistent. Therefore, it is reasonable to represent the atmospheric aerosol concentrations using WRF-Chem modeling.

In this study, we selected three fire sites to evaluate the morphological effects on the BC ARI. The positions of the selected sites are shown in Table S5, and they represent the fire sites with high aerosol concentrations. As shown in Figure 3, the temporal BC concentrations at fire sites can even exceed approximately $400\ \mu g/m^3$ when the fire occurs, while the BC concentrations are extremely low in other days. As shown in Figure 3, even though the maximum BC concentrations in the polluted urban cities are much smaller than the fire sites, the mean BC concentrations can reach approximately $12\ \mu g/m^3$. The simulated BC concentrations generally agree with the measurements of Zhang et al. (2012) for the urban region, where BC concentrations were observed to be approximately $4 - 12.7\ \mu g/m^3$. We further compare the calculated AOD and AAOD with observations from AErosol RObotic NETwork (AERONET). The AERONET data of Beijing is available among all selected cities in Eastern China. As shown in Figure 4, the calculated AAOD and AOD can generally represent the observations.

Figure 5 shows the AOD of BC with different morphologies, where BC AOD was calculated by the difference between AOD with BC and that without BC. Zhuang et al. (2019) indicated that the time-averaged BC AOD in Beijing, Hefei, and

235 Taihu was approximately 0.05 – 0.06. Our simulations are generally in agreement with their results, and the simulated BC AOD varies from approximately 0.01 to approximately 0.12 in the selected urban cities. In Beijing and Tianjin, BC AOD can reach approximately 0.12, while in Shanghai and Nanjing, the maximum BC AOD is approximately 0.07 and 0.1, respectively. The maximum BC AOD can reach 0.5 – 0.9 in typical fire sites. From Figure 5, we can also see that BC AOD calculated using the spherical model is relatively higher than those using fractal aggregate models, which is consistent with the findings of Liu

and Mishchenko (2005). As shown in Figure 6, in the urban area a spherical assumption for BC lead to an overestimation of less than 0.03 for AOD compared to the fractal model with a $D_f$ of 1.8, while the overestimation can reach approximately 0.15 in fire sites. The overestimation accounts for a large proportion of BC AOD, which can exceed 20% of the total BC AOD.

Figure 7 shows the calculated BC AAOD using different BC models, where BC AAOD was calculated by the difference between AAOD with BC and that without BC. . Our modeling results show that a more compact structure may lead to a smaller

AAOD, and this is consistent with the findings of Liu and Mishchenko (2005) for single BC particles. The reason is that a more compact structure blocks the light from transmitting into the inner part of the particle, and a smaller absorption efficiency was observed (Kahnert and Devasthale, 2011). Shin et al. (2019a) showed that most BC AAOD at 440 nm in Asia is within the range of 0 – 0.12. Our simulated BC AAOD at 450 nm is generally consistent with their findings. At fire sites, BC AAOD at 450 nm wavelength can reach approximately 0.7. As shown in Figure 8, the spherical model underestimates AAOD by approximately

0.016 compared to the fractal model with a $D_f$ of 1.8 in typical polluted cities, while the AAOD underestimation using the spherical model can reach approximately 0.04 in fire sites. In general, the AAOD underestimation using the spherical model is approximately 8% of the total BC AAOD.

Figure 9 shows the time-averaged BC AOD at 550 nm and AAOD at 450 nm as a percentage of the total AOD and AAOD, respectively. In typical China polluted cities, BC AOD at 550 nm accounts for approximately 4.6% – 7% of the total AOD,

while BC AOD in fire sites can account for larger than 10% of the total AOD. At 450 nm, in both polluted urban and fire sites, the fractions of BC AAOD are close, which is approximately 30%. This means that the relative proportions of BC and OC in polluted urban sites are close to those of fire sites.

BC morphologies also have significant impacts on the BC AOD and AAOD fractions. As BC morphologies change from a fractal dimension of 1.8 to a spherical structure, BC AOD fraction increases from 4.6% – 5.5% to 5.8% – 6.9% in typical

polluted urban cities and from 9.0% – 10.3% to 11.1% – 12.7% in typical fire sites, respectively. The relative differences between the fractal model and the spherical model ((Fractal – Sphere)/Sphere) can be above 25%. For BC AAOD fraction, the values can vary in the range of 25.0% – 33.2% in typical polluted urban cities and the range of 25.4% – 30.0% in typical fire sites, respectively. The AAOD fraction relative differences between the fractal model with a $D_f$ of 1.8 and the spherical model are approximately 10%.

**3.2  Impacts of BC morphology on SSA**

SSA, as the ratio of scattering to extinction, is widely used to infer aerosol types. Figure 10 shows the comparison of SSA using different BC models. Shin et al. (2019b) showed that the mean SSA at 440 nm was approximately 0.89 – 0.92 in Beijing using AERONET data, and Shaheen et al. (2019) demonstrated that the mean SSA at 440 nm was approximately 0.897 in

winter Beijing. At 450 nm, the calculated SSA in typical urban cities is within the range of a 0.86 – 0.92, which is generally consistent with the observations in previous studies. While in the fire region, SSA varies in a wider range. At 450 nm, SSA at the selected fire sites can vary in the range of 0.75 – 0.9, which is a little smaller than that in polluted urban areas due to large portions of carbonaceous aerosols in the fire region.

With more compact structures, SSA presents a larger value, which is consistent with the findings of Kahnert and Devasthale (2011). However, the effects of BC morphologies on total SSA at 450 nm are not obvious due to the small percentage of BC in the atmosphere. As shown in Figure 11, the overestimations of the spherical BC model for SSA at 450 nm are generally within 0.005 at typical polluted cities and within 0.012 at fire sites, which is less than 1% of the total SSA. However, these values may have relatively larger impacts when evaluating the climate effect of BC, as BC commonly presents a relatively small value of 0.2 – 0.4 (Kahnert and Devasthale, 2011).

### 3.3 Impacts of BC morphology on ARI

BC clear-sky ARI varies in different regions and the reported BC ARI varies in previous studies. Zhuang et al. (2018) estimated BC clear-sky ARI to be +1.85 $Wm^{-2}$ in East Asia. Much larger BC clear-sky ARI during December (+15 $Wm^{-2}$) and November (+8 $Wm^{-2}$) over Ahmedabad and Gurushikhar, respectively, were reported by Rajesh and Ramachandran (2018). Zhuang et al. (2019) showed that clear-sky ARI averaged over East Asia were +0.02 to +1.34 $W/m^2$ in summer in eastern Asia. Lu et al. (2020) showed daily-mean BC clear-sky ARI were within the range of +1.37 – 4.89 $Wm^{-2}$ in Beijing. Our calculated daily-mean BC ARI in winter generally agrees with those previous studies. The daily-mean clear-sky BC ARI at the TOA in typical sites using different BC models is presented in Figure 12. In winter, BC clear-sky ARI at typically polluted cities varies in the range of approximately +0.5 – 5.0 $Wm^{-2}$. While in large fire sites, the daily mean BC clear-sky ARI exceeds +8.0 $Wm^{-2}$. Generally, with a more compact structure, BC presents a smaller clear-sky ARI at the TOA. The reason can be explained from the following aspects. Fractal BC can more efficiently absorb than spherical BC, while the total scattering is not significantly modified. Thus, the fractal BC leads to a larger positive clear-sky ARI.

**Table 1.** Time-averaged BC Clear-sky ARI at different sites ($Wm^{-2}$).

| Location | $D_f$=1.8 | $D_f$=2.2 | $D_f$=2.6 | Sphere |
|----------|-----------|-----------|-----------|--------|
| Beijing | +1.76 | +1.68 | +1.63 | +1.57 |
| Shanghai | +1.52 | +1.45 | +1.38 | +1.26 |
| Tianjin | +2.00 | +1.91 | +1.86 | +1.77 |
| Nanjing | +1.99 | +1.90 | +1.83 | +1.70 |
| Fire Loc1 | +5.39 | +5.16 | +5.00 | +4.69 |
| Fire Loc2 | +8.60 | +8.27 | +8.12 | +7.78 |
| Fire Loc3 | +5.61 | +5.38 | +5.22 | +4.91 |

Table 1 shows the time-averaged BC clear-sky ARI (averaging BC clear-sky ARI over the simulation period) in typical polluted urban sites and fire sites. Using the spherical model, the time-averaged clear-sky ARI is +1.26 – +1.77 $Wm^{-2}$ in

**Table 2.** Time-averaged BC All-sky ARI at different sites (Wm$^{-2}$).

| Location | $D_f$=1.8 | $D_f$=2.2 | $D_f$=2.6 | Sphere |
|----------|-----------|-----------|-----------|--------|
| Beijing | +1.77 | +1.70 | +1.68 | +1.67 |
| Shanghai | +2.26 | +2.18 | +2.15 | +2.15 |
| Tianjin | +2.00 | +1.93 | +1.90 | +1.88 |
| Nanjing | +2.79 | +2.69 | +2.66 | +2.66 |
| Fire Loc1 | +5.14 | +4.93 | +4.81 | +4.62 |
| Fire Loc2 | +7.38 | +7.11 | +6.99 | +6.77 |
| Fire Loc3 | +5.22 | +5.02 | +4.89 | +4.69 |

typical polluted urban cities, while it increases to +1.52 – +2.00 Wm$^{-2}$ using a fractal aggregate model with a $D_f$ of 1.8. The relative differences between the fractal model with a $D_f$ of 1.8 and spherical model ((Fractal – Sphere)/Sphere) can reach approximately 12.1% – 20.6% in typical urban cities. In fire sites, when modifying BC structure from a sphere to a fractal aggregate with a $D_f$ of 1.8, the time-averaged BC clear-sky ARI increases from +4.69 – +7.78 Wm$^{-2}$, to +5.39 – +8.60 Wm$^{-2}$, and the relative differences between the two models are 10.5% – 14.9%. Lu et al. (2020) showed that BC shapes can introduce approximately 5% relative uncertainties in eastern China using different measured BC profiles. However, our results show that much larger uncertainties can be introduced from BC morphologies. The reason is that $D_f = 2.8$ was assumed for BC aggregates in the study of Lu et al. (2020), which are close to spherical shape. This $D_f$ value is larger than the observed $D_f$. Besides, due to different solar zenith angles, our results show the ARI uncertainties caused by BC morphologies may vary in different regions. Therefore, the BC morphological effects on the BC ARI should be carefully considered in different regions.

IPCC (2014) has reported a global mean BC all-sky ARI of +0.6 W m$^{-2}$ , while Wang et al. (2014) estimated a smaller all-sky ARI of +0.13 W m$^{-2}$ based on the constraints from the mass and absorption observations. Tuccella et al. (2020b) further showed that the global mean BC all-sky ARI is in the range of +0.13 and +0.25 W m$^{-2}$. However, the BC all-sky ARI in some specific regions is relatively large. Based on in-situ measurements, Lamb et al. (2018) have estimated the mean column all-sky BC ARI to be +0.48 to +2.01 Wm$^{-2}$ over South Korea. In Beijing, Sun et al. (2022) found that the mean BC all-sky ARI decreased from +3.36 W m$^{-2}$ in 2012 to +1.09 W m$^{-2}$ in 2020. The daily-mean all-sky ARI in typical urban polluted cities in eastern China and fire sites in the northwest US are shown in Figure 13. The daily-mean all-sky ARI estimated in this study generally varies in the range of approximately +0.2 – +4.5 Wm$^{-2}$ in typical polluted cities. Table 2 shows the time-averaged BC all-sky ARI (averaging BC all-sky ARI over the simulation period) in different sites. The time-averaged all-sky ARI are +1.67 – +2.79 Wm$^{-2}$ in urban cities, and it is relatively high compared to the regional-mean all-sky ARI due to the high BC emissions. Our estimated BC all-sky ARI is generally in the range reported by Sun et al. (2022).

The differences of all-sky ARI between the fractal model and spherical model are smaller than those of clear-sky ARI in typical polluted cities in eastern China. The relative differences in the time-averaged all-sky ARI between fractal model and the spherical model are below 6.5% in typical polluted cities. The all-sky ARI at fire sites in the northwest US is smaller than

the clear-sky ARI. The time-averaged all-sky ARI are $+4.62 - +7.38$ Wm$^{-2}$ in typical fire sites. The relative differences in the time-averaged all-sky ARI between the fractal model with a $D_f$ of 1.8 and the spherical model are $9.0\% - 11.3\%$ in typical fire sites, which is relatively smaller than the differences for clear-sky ARI.

The regional-mean ARI are shown in Figures 14 – 15. As shown in Figure 14, the BC clear-sky ARI of exceeding $+3.0$ W/m$^2$ is observed in eastern China. With a spherical BC model, the regional-mean clear-sky ARI in eastern China is estimated as $+1.35$ Wm$^{-2}$, and it agrees well with the reported regional-mean clear-sky ARI of $+1.34$ Wm$^{-2}$ in East Asia by Zhuang et al. (2019). BC morphologies also have a non-ignorable impact on the BC ARI. The regional-mean ARI deviations between the fractal model and spherical model are $+0.23$, $+0.15$, and $+0.1$ Wm$^{-2}$ when $D_f$ is 1.8, 2.2, and 2.6, respectively, and the relative differences are $17.1\%$, $11.1\%$, and $7.4\%$, respectively.

The BC morphologies have a relatively small impact on the all-sky ARI. As shown in the lower panels of Figure 14, the all-sky ARI is generally larger than the clear-sky ARI, while the deviations between the fractal model and spherical model are smaller compared to those for clear-sky. The BC all-sky ARI can exceed $+4.0$ W/m$^2$ in eastern China. A relatively larger regional-mean all-sky ARI is observed than clear-sky ARI, which is $+1.79$ Wm$^{-2}$ in eastern China when using the spherical model. The regional-mean all-sky ARI differences between the fractal model and spherical model are $0.13$, $0.06$, $0.03$ Wm$^{-2}$ when $D_f$ is 1.8, 2.2, and 2.6, respectively. The relative differences for all-sky ARI between the two models are less than $7.3\%$.

The regional-mean clear-sky ARI in the fire region in the northwest US is shown in Figure 15. The clear-sky ARI in the fire sites is obviously larger than the other sites. The BC clear-sky ARI of exceeding $+5.0$ W/m$^2$ is observed, and the regional-mean clear-sky ARI is $+0.93$ W/m$^2$. The differences of regional-mean clear-sky ARI between the fractal model and the spherical model in the fire region in the northwest US are more substantial than those in eastern China, which reach approximately $+0.36$, $+0.29$, and $+0.19$ W/m$^2$ when $D_f$ is 1.8, 2.2, and 2.6, respectively. The relative differences between the fractal model and spherical model are $38.7\%$, $31.2\%$, and $20.4\%$ when $D_f$ is 1.8, 2.2, and 2.6, respectively. Smaller all-sky ARI than clear-sky ARI are observed at the sites where fire occurs, while the regional-mean all-sky ARI in the fire region in the northwest US is generally larger than clear-sky ARI, which is $+1.67$ W/m$^2$. The differences between the two models are $+0.28$, $+0.19$, and $+0.11$ W/m$^2$ when $D_f$ is 1.8, 2.2, and 2.6, respectively, and the relative differences are $16.8\%$, $11.4\%$, and $6.6\%$.

## 4   Discussion

In current climate models, such as CESM, MIROC-SPRINTARS, and WRF-Chem, the Mie theory was commonly used to calculate the optical properties of BC aerosols. However, fractal-like BC aerosols were often observed in the atmosphere. In this work, we found that the effects of BC morphology are spatially-dependent. Compared to the spherical BC model, the fractal BC model generally presents a larger clear-sky ARI, which may lead to the underestimations of BC ARI in the climate models. The relative differences in the time-averaged clear-sky ARI are $12.1\% - 20.6\%$ and $10.5\% - 14.9\%$ in typical polluted urban cities and fire sites, respectively. Furthermore, the regional-mean clear-sky ARI is also significantly affected by the BC morphology, and relative differences of $17.1\%$ and $38.7\%$ between the fractal model were observed in eastern China and in the northwest US, respectively, while the existence of cloud would weaken the BC morphological effects. The results imply that

current climate modeling may significantly underestimate the BC ARI uncertainties as the morphological effects on BC ARI are ignored in most climate models.

However, this work is by no means exhaustive. This work assumed that BC aerosols are externally mixed with other chemical components, while BC aerosols are often internally mixed with other components, such as organic aerosols, sulfate, etc (China et al., 2013a; Adachi et al., 2010; Wang et al., 2021b). BC absorption can be significantly enhanced by the "lensing Effect" even if BC aerosols are internally mixed with non-absorbing materials, which may lead to larger BC ARI (Chung et al., 2012; Liu et al., 2017a). Previous studies have shown that the morphologies of internally mixed BC would significantly affect its absorption enhancement (Luo et al., 2019; Wang et al., 2021a; Luo et al., 2021c), so lead to larger uncertainties in the estimation of BC ARI. Thus, the sensitivities of BC morphologies on the ARI estimated in this work may be smaller than those in real cases.

Futhermore, we found that the spherical assumption generally unerestimates the clear-sky ARI for externally mixed BC, while oppsite phenomenon may be found for internally mixed BC. A core-shell spherical morphology was widely used to represent the internally mixed BC. However, many partially coated BC aerosols exist in the atmosphere, while the core-shell spherical BC model commonly assumes the BC is fully embedded in a coating shell. The core-shell morphology may overestimate the absorption of partially coated BC (Wang et al., 2021a; Zhang et al., 2018), so overestimate the ARI. Thus, the ARI of internally mixed BC with complex morphologies should be further investigated in the future.

## 5 Summary and Conclusions

The current climate modeling commonly assumes a spherical morphology for BC, while the fractal structure is more realistic than the spherical morphology for externally mixed BC. In this work, we used the fractal model to re-evaluate the BC ARI in a typical polluted urban area in eastern China and a fire region in the northwest US. We found that BC morphologies have non-ignorable impacts on the aerosol optical properties. At 550 nm wavelength, the spherical BC model can overestimate the AOD up to 0.03 and 0.15 in typical polluted cities in China and fire sites in US, respectively. The overestimations roughly account for 20% of the total BC AOD. Besides, the spherical BC model underestimates BC AAOD at 450 nm up to 0.016 and 0.04 at typical polluted cities in China and fire sites in the US, respectively, compared to the fractal model with a $D_f$ of 1.8. The underestimations account for approximately 8% of the total BC AAOD.

Both the morphological effects on clear-sky and all-sky ARI are evaluated. With a spherical BC model, the estimated time-averaged clear-sky ARI is generally in the range of +1.26 – +1.77 W/m$^2$ in typical urban polluted cities in eastern China, while this range increases to approximately +1.52 – +2.00 W/m$^2$ when using the fractal model with a $D_f$ of 1.8. The clear-sky ARI relative differences between the two models are approximately 12.1% – 20.6% in typical urban polluted cities. In fire sites, when modifying BC structure from a sphere to a fractal aggregate, the time-averaged BC clear-sky ARI increases from +4.69 – +7.78 Wm$^{-2}$ to +5.39 – +8.60 Wm$^{-2}$ in typical fire sites. The relative differences between the two models are approximately 10.5% – 14.9% in typical fire sites. The existence of clouds weaken the effects of BC morphologies on the ARI. The all-sky

ARI relative differences between the fractal model and the spherical model are approximately 4.9% – 6.4% and 9.0% – 11.3% in typical urban polluted cities and typical fire sites, respectively, which is relatively smaller than those of clear-sky ARI.

The impacts of BC morphologies on the regional-mean ARI were also evaluated. The regional-mean clear-sky ARI was estimated as +1.35 $Wm^{-2}$ and +0.93 $Wm^{-2}$ in the polluted urban area and the fire region, respectively, using the spherical BC model. The regional-mean clear-sky ARI differences between the fractal model and the spherical model are approximately +0.23 and + 0.36 $Wm^{-2}$ in these two regions, respectively, and the relative differences between the two models are approximately 17.1% and 38.7%, respectively. The all-sky ARI differences between the fractal model and the spherical model are

relatively smaller. The relative differences in the regional-mean all-sky ARI between the fractal model and the spherical model are less than 7.3% and 16.8% in these two regions, respectively. Thus, the effects of BC morphologies on the ARI should be carefully considered in different regions.

*Acknowledgements.* This work was financially supported by the National Natural Science Foundation of China (Grant No. 41925019), the National Key Research and Development Plan under Grant No. 2020YFC1511600, and the Fundamental Research Funds for the Central Universities under Grant No. WK2320000052. RKC and CZ acknowledge support from the US National Science Foundation (AGS-1926817) and the NASA ACCDAM program (NNH20ZDA001N). CZ would like to acknowledge partial support received from the McDonnell International Scholars Academy at Washington University in St. Louis.).

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

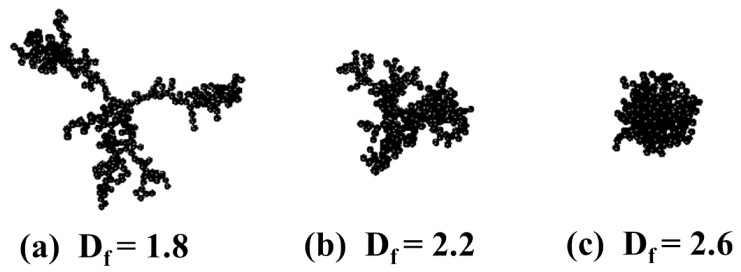

**(a)** $D_f = 1.8$     **(b)** $D_f = 2.2$     **(c)** $D_f = 2.6$

**Figure 1.** Typical morphologies of fractal BC.

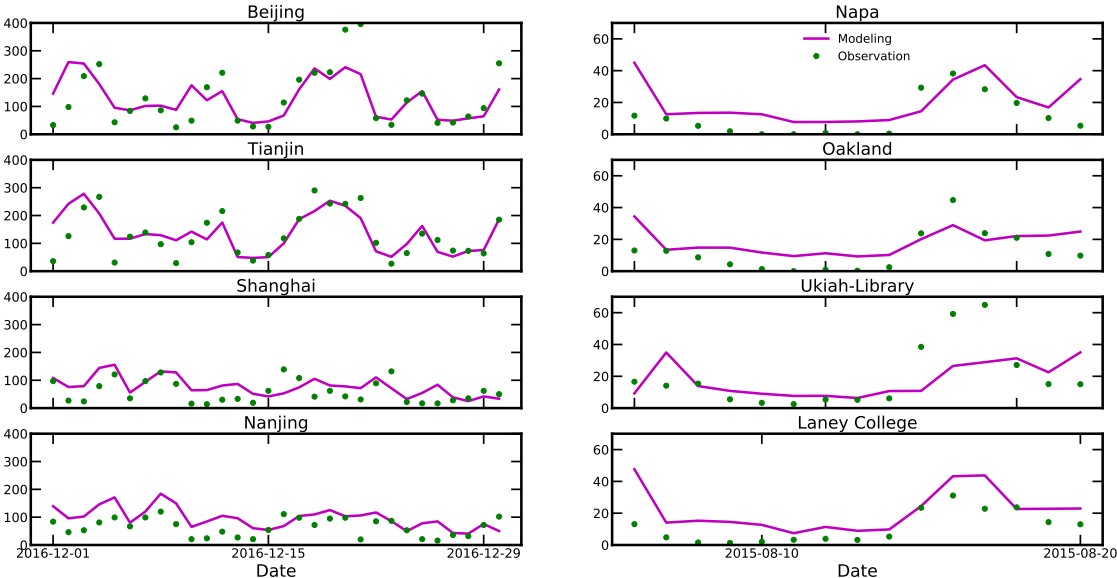

**Figure 2.** Comparison of measured and calculated PM2.5 concentrations.

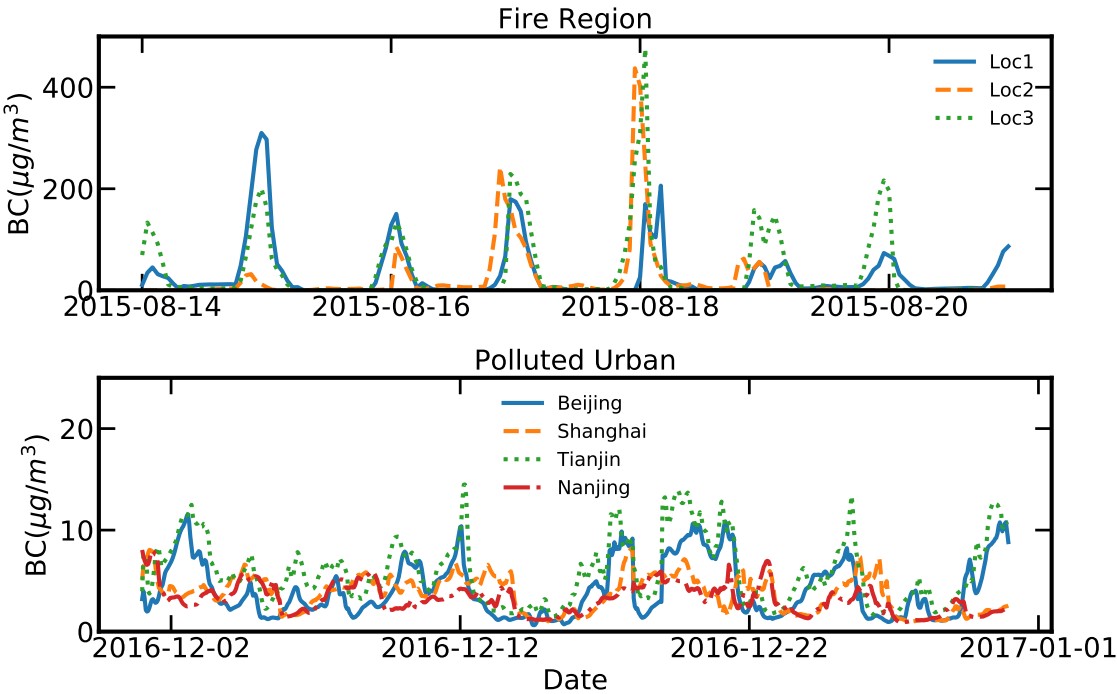

**Figure 3.** The time series of BC concentrations in typical northwest US and eastern China.

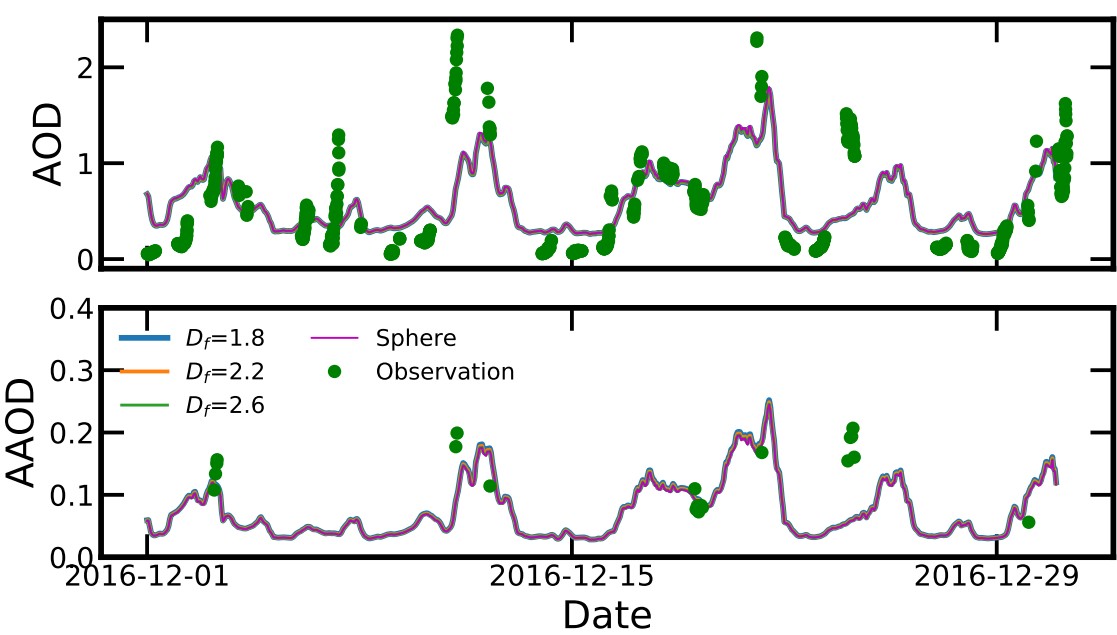

**Figure 4.** Comparison of measured and calculated AOD and AAOD in Beijing, where λ=500 nm for AOD. λ=440 nm and 450 nm for measured and calculated AAOD, respectively, and the observations were obtained from AERONET.

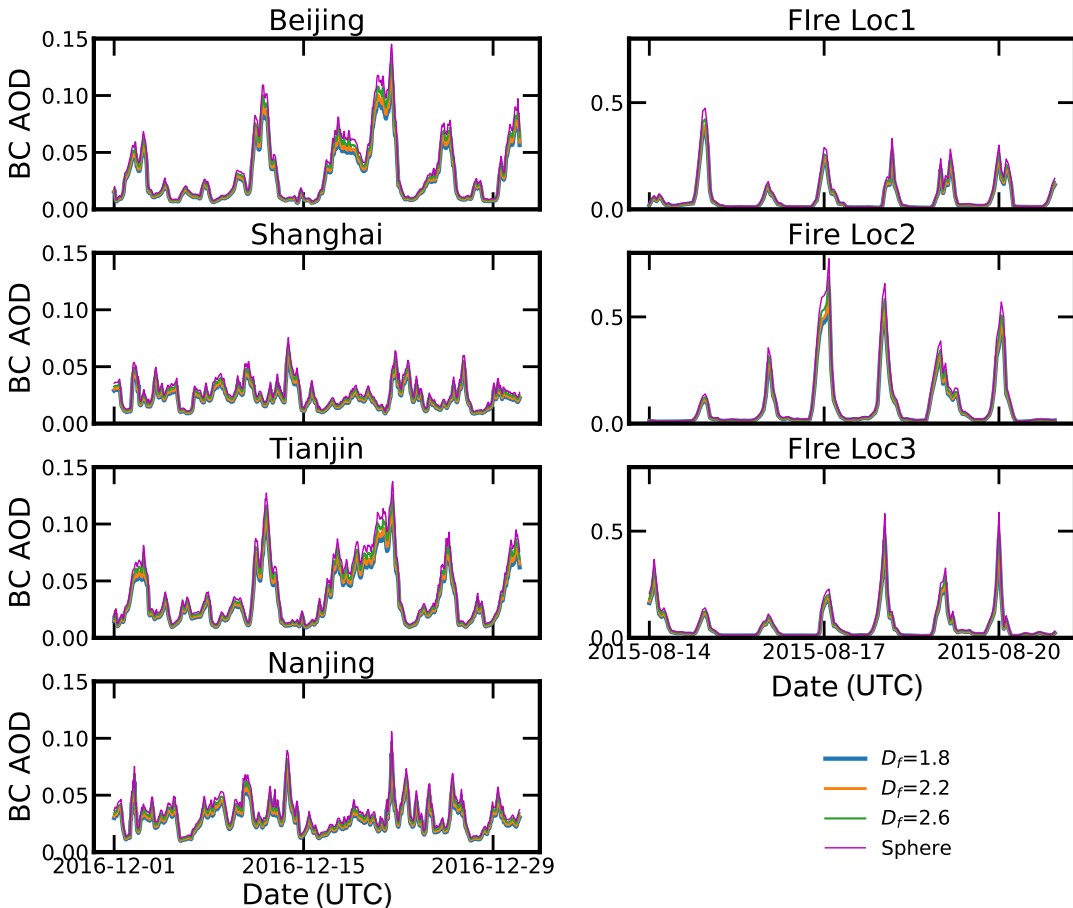

**Figure 5.** The comparison of BC AOD for different BC morphologies, λ=550 nm.

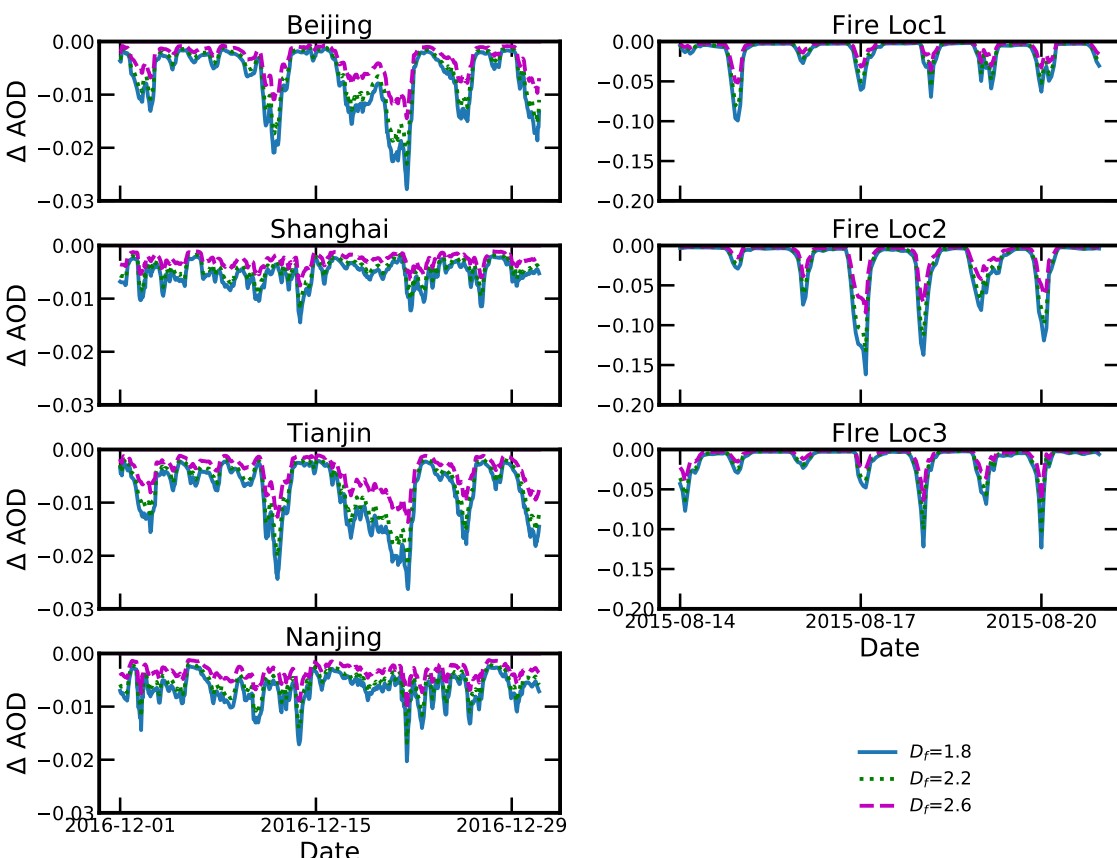

**Figure 6.** The AOD difference between fractal aggregate models and the spherical model, λ=550 nm.

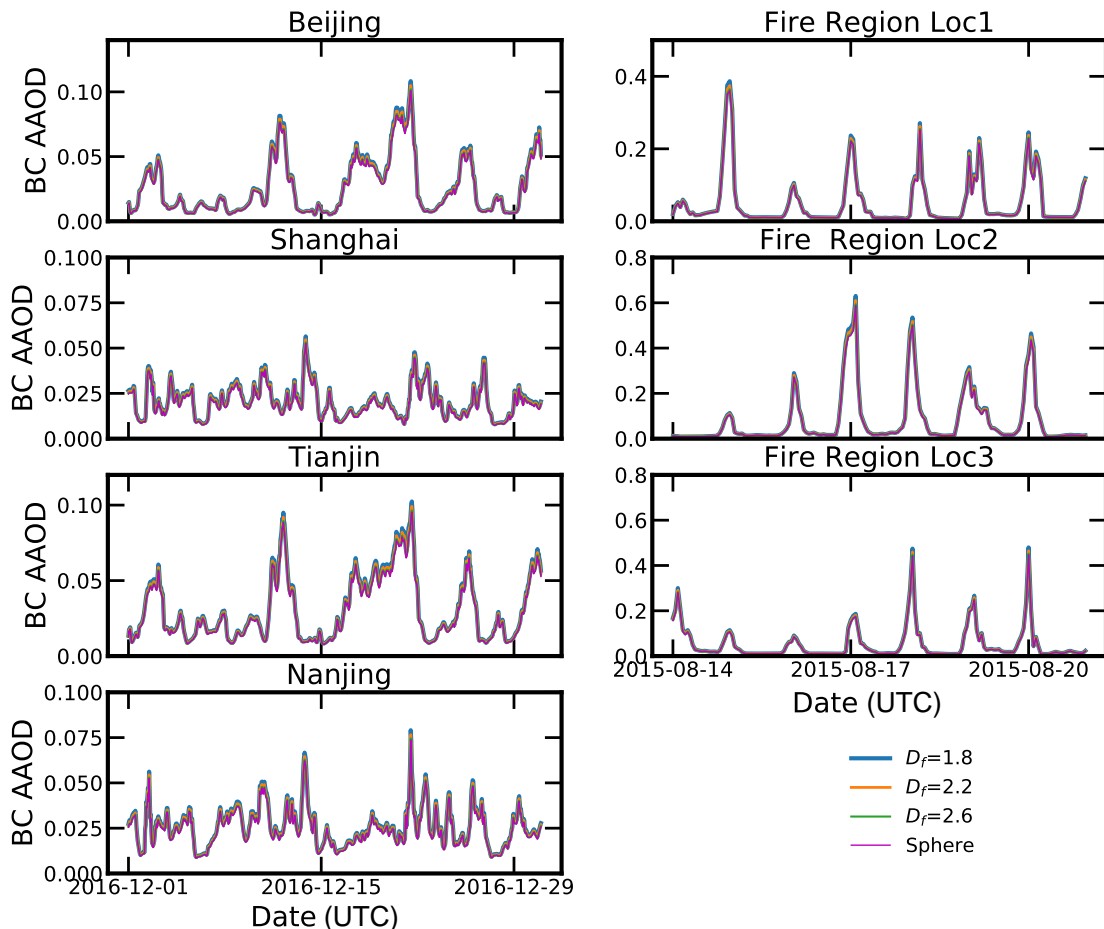

**Figure 7.** The comparison of BC AAOD for different BC morphologies, $\lambda$=450 nm.

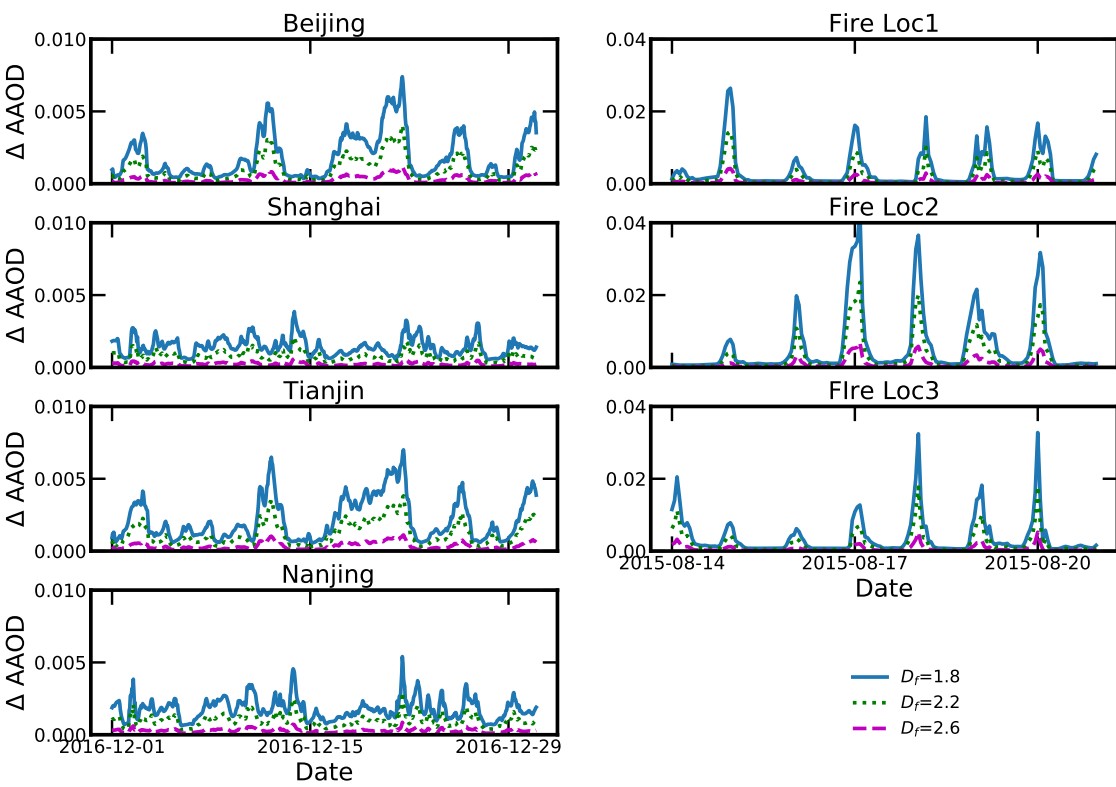

**Figure 8.** The AAOD difference between fractal aggregate models and the spherical model, $\lambda = 450$ nm.

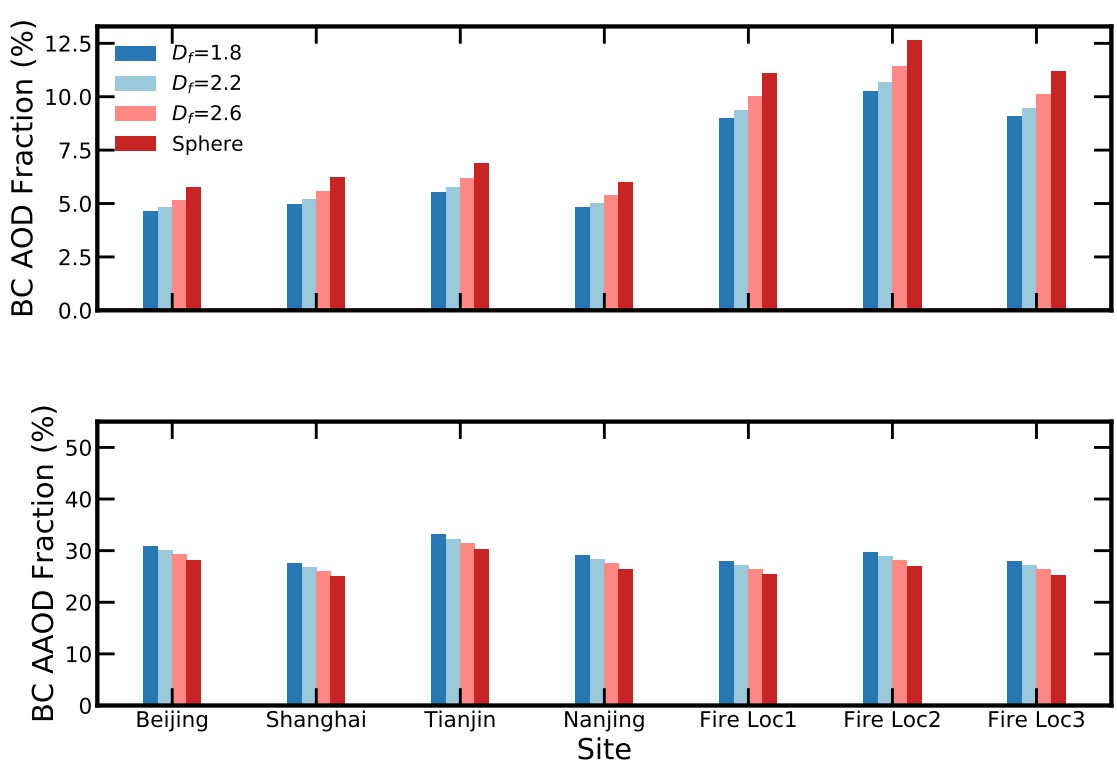

**Figure 9.** BC AOD (550 nm) and AAOD(450 nm) as a percentage of total AOD and AAOD.

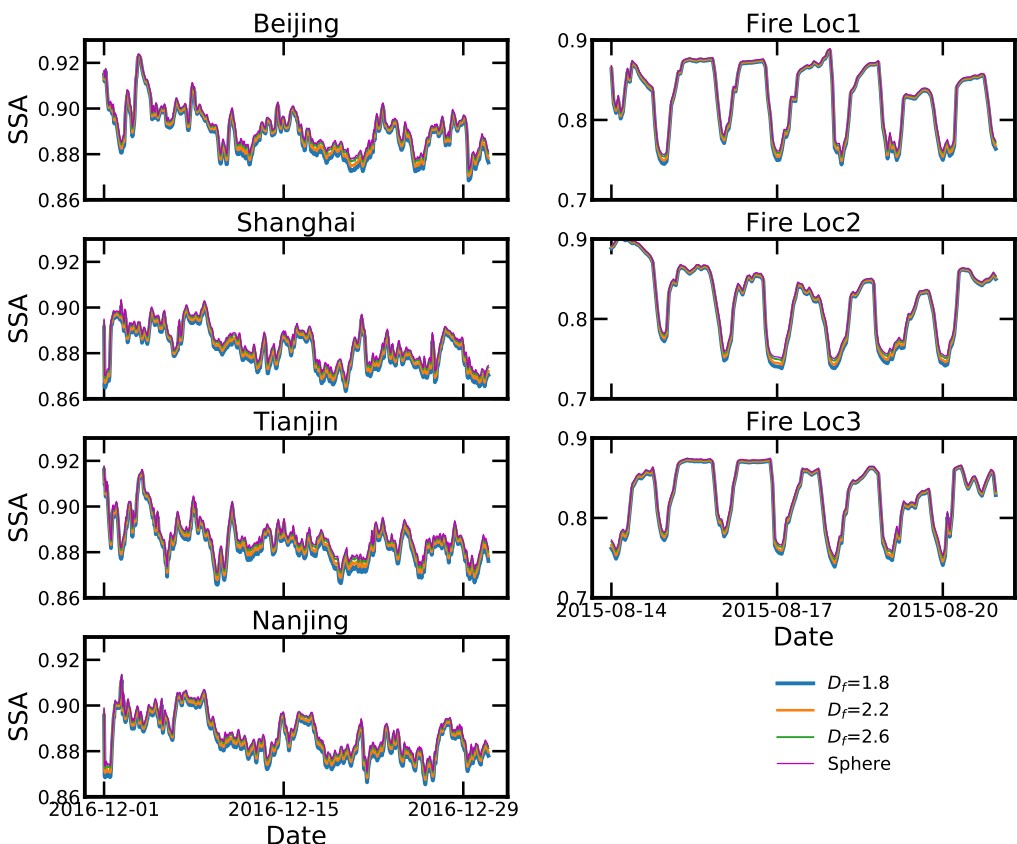

**Figure 10.** The comparison of SSA using different BC models, λ=450 nm.

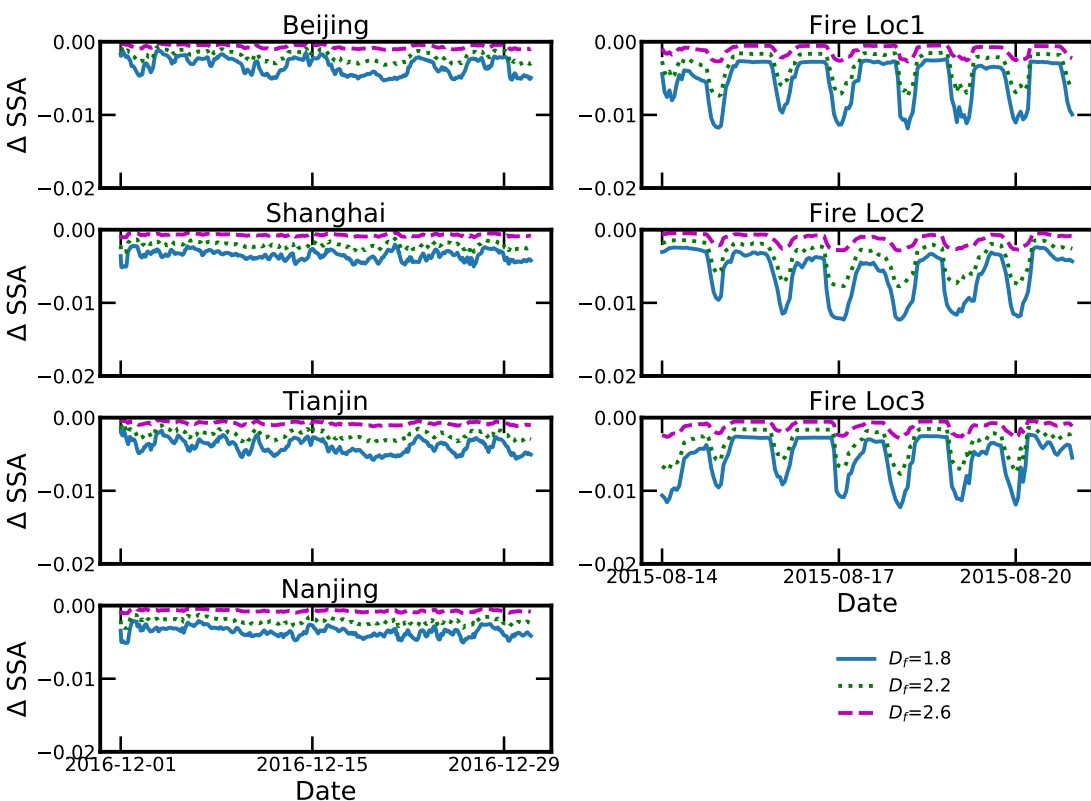

**Figure 11.** The SSA differences between fractal aggregate models and the spherical model, $\lambda$=450 nm.

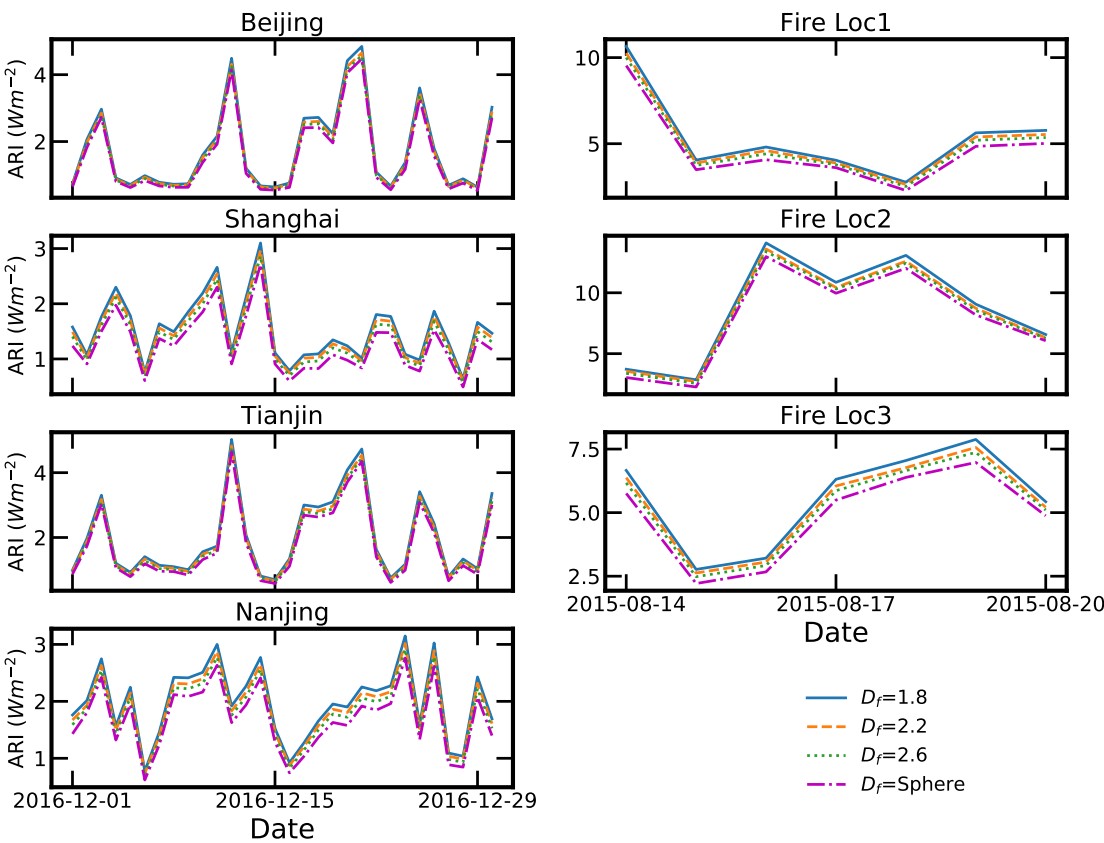

**Figure 12.** The clear-sky BC ARI at the typical urban polluted cities in eastern China and fire sites in the northwest US calculated using different BC models.

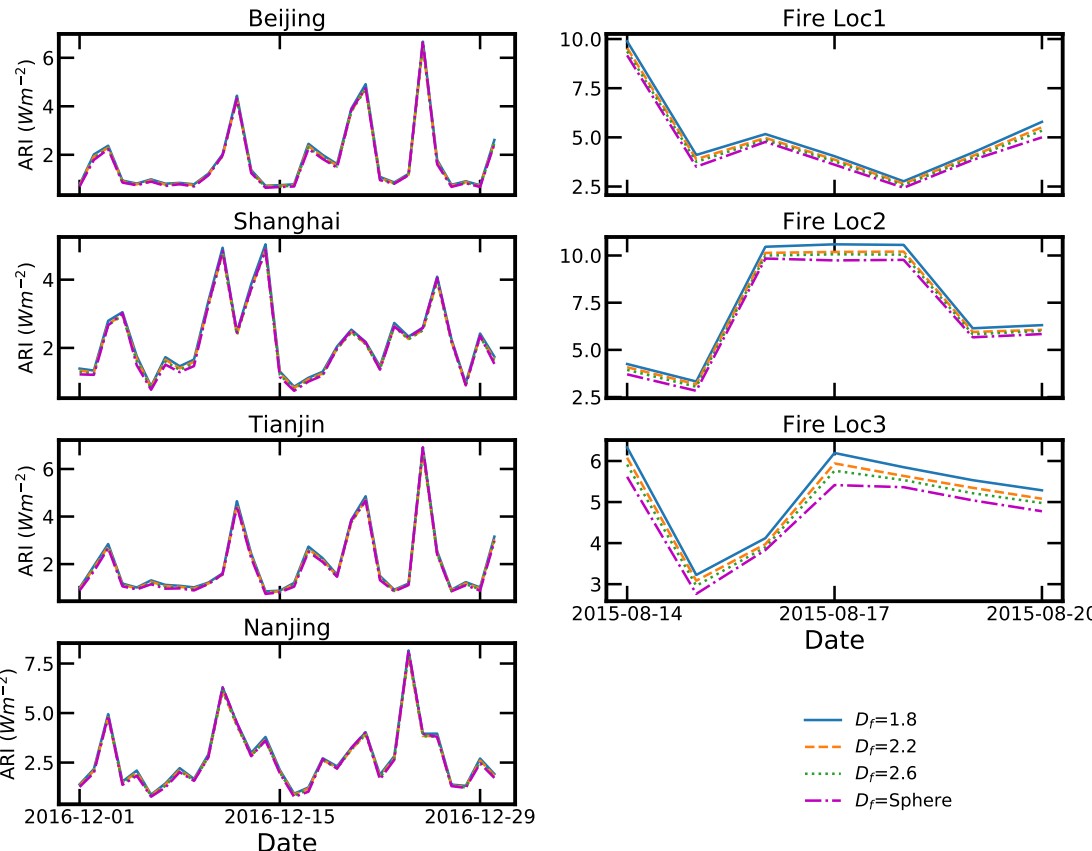

**Figure 13.** The all-sky BC ARI at the typical urban polluted cities in eastern China and fire sites in the northwest US calculated using different BC models..

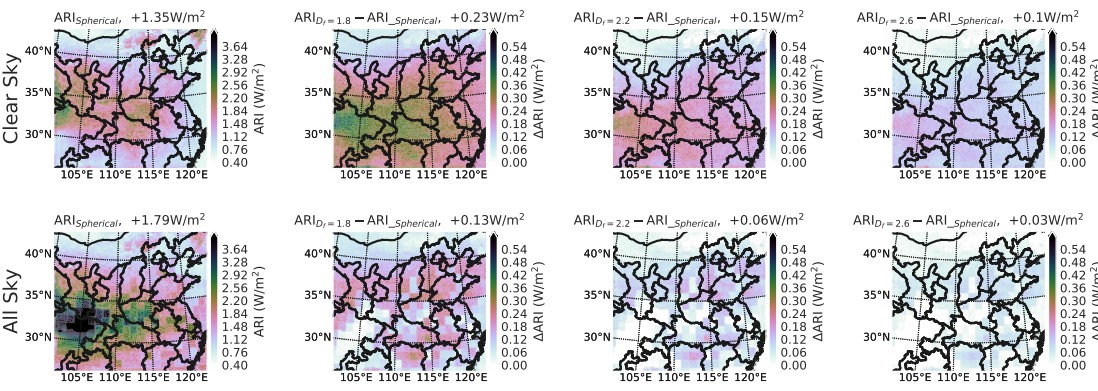

**Figure 14.** The regional-mean BC ARI at the TOA in eastern China for different particle models.

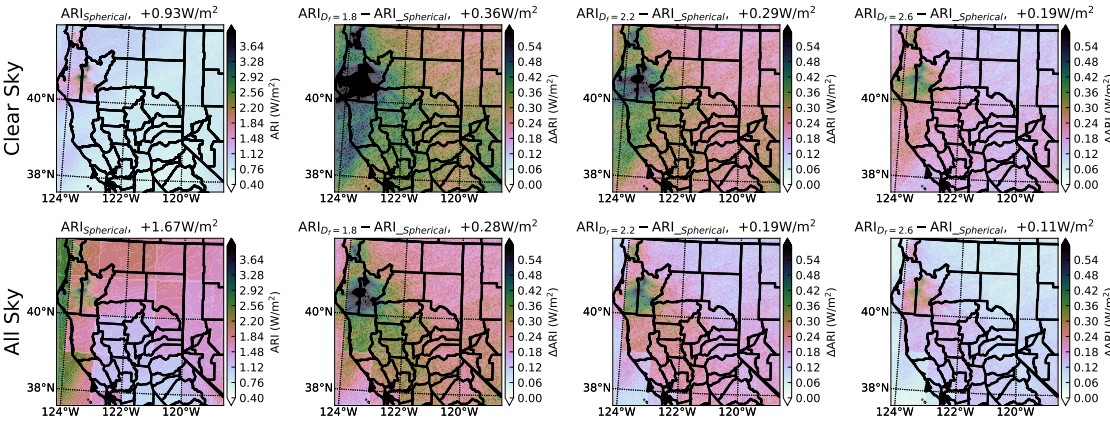

**Figure 15.** Similar to Figure 14, but for northwest US.

*Code availability.* The WRF-Chem is publicly available from https://ruc.noaa.gov/wrf/wrf-chem/; MSTM can be download from https://www.eng.auburn.ckwski/scatcodes/; The libRadtran is available from http://www.libradtran.org/doku.php; The FlexAOD can be requested from Prof. Curci (http://pumpkin.aquila.infn.it/flexaod/).

*Data availability.* The look-up tables calculated in this work can be obtained from https://figshare.com/articles/dataset/Look_up_tables_zip/13096241.
The PM2.5 data in China was obtained from https://www.aqistudy.cn/historydata/, and PM2.5 data in the northwest US can be found from
https://www.epa.gov/outdoor-air-quality-data/download-daily-data.

*Author contributions.* JL and QXZ conceived the presented idea. JL developed the models, performed the computations, and wrote the paper.
ZQL, CZ, YMZ, RKC, YZ verified the simulation methods and results. QXZ revised the paper and supervised the findings of this work. GC
developed the FlexAOD model. All authors discussed the results and contributed to the final paper.

*Competing interests.* The authors declare that they have no conflict of interest.