# Peer review of "Regional impacts of black carbon morphologies on shortwave aerosol-radiation interactions: A comparative study between the US and China"

_Atmospheric Chemistry and Physics, 2021_

## Author Comment (AC1)

**Response to the comments of Reviewer #1**

First of all, we would like to thank the two anonymous reviewers for their thoughtful reviews and valuable comments on the manuscript. In the revision, we have accommodated all the suggested changes into consideration and revised the manuscript accordingly. All changes are highlighted in the revised manuscript in **BLUE** in the revision. In this response, the questions and comments of reviewers are in **BLACK** font, and responses are highlighted in **BLUE.** The changes made in the revised manuscript are marked in **RED** font.

The authors investigated ARI of BC and some other radiative properties with a modified BC-model, as the fractal-like structure is more realistic. The authors found some biases in the traditional sphere BC models in AOD, AAOD and AAE. For ARI, the more realistic BC model produces higher ARI forcing compared with sphere model. The method of this study is solid and the results are well presented, providing some new insights of BC ARI to the community.

However, this paper needs some revisions before the acceptance for publication.

Major comments:

Comments: It looks like a *discussion* section is missing. A more detailed discussion is needed (e.g., limitations, interpretation of the results, comparison with previous studies).

**Response:** Thanks very much for your comments. In the revised manuscript, a discussion section is added, and the contents are shown in follows:

"In current climate models, such as CESM, MIROC-SPRINTARS, and WRF-Chem, the Mie theory was commonly used to calculate the optical properties of BC aerosols. However, fractal-like BC aerosols were often observed in the atmosphere. In this work, we found that the effects of BC morphology are spatially-dependent. Compared to the spherical BC model, the fractal BC model generally presents a larger clear-sky ARI, which may lead to the underestimations of BC ARI in the climate models. The relative differences in the time-averaged clear-sky ARI are 12.1% – 20.6% and 10.5% – 14.9% in typical polluted urban cities and fre sites, respectively. Furthermore, the regional-mean clear-sky ARI is also signifcantly affected by the BC morphology, and relative differences of 17.1% and 38.7% between the fractal model were observed in eastern China and in the northwest US, respectively, while the existence of cloud would weaken the BC morphological effects. The results imply that current climate modeling may signifcantly underestimate the BC ARI uncertainties as the morphological effects on BC ARI are ignored in most climate models. However, this work is by no means

exhaustive. This work assumed that BC aerosols are externally mixed with other chemical components, while BC aerosols are often internally mixed with other components, such as organic aerosols, sulfate, etc (China et al., 2013a; Adachi et al., 2010; Wang et al., 2021b). BC absorption can be signifcantly enhanced by the "lensing Effect" even if BC aerosols are internally mixed with non-absorbing materials, which may lead to larger BC ARI (Chung et al., 2012; Liu et al., 2017a). Previous studies have shown that the morphologies of internally mixed BC would signifcantly affect its absorption enhancement (Luo et al., 2019; Wang et al., 2021a; Luo et al., 2021c), so lead to larger uncertainties in the estimation of BC ARI. Thus, the sensitivies of BC morphologies on the ARI estimated in this work may be smaller than those in real cases.

Futhermore, we found that the spherical assumption generally unerestimates the clear-sky ARI for externally mixed BC, hile oppsite phenomenon may be found for internally mixed BC. A core-shell spherical morphology was widely used to epresent the internally mixed BC. However, many partially coated BC aerosols exist in the atmosphere, while the coreshell spherical BC model commonly assumes the BC is fully embedded in a coating shell. The core-shell morphology may verestimate the absorption of partially coated BC (Wang et al., 2021a; Zhang et al., 2018), so overestimate the ARI. Thus, the RI of internally mixed BC with complex morphologies should be further investigated in the future.

"

**Comments:** I was wondering in the current generation of GCMs or ESMs such as CMIP5/6, is the BC model sphere or fractal-like? Or maybe some of them are sphere/fractal. I suggest authors provide some more information on this in the *introduction* section. If most of the current models use simplified sphere BC models, then the contributions of this study would be more significant and the authors should add some discussions in the *discussion* section.

**Response:** Thanks very much for your comments. In the current models, the spherical BC model is used, and we have added some descriptions in the introduction:

An important cause of the discrepancy is BC's complex morphology. BC aerosols are assumed as spheres, and the optical properties are calculated using the Mie theory in most climate and atmospheric chemical transport models, such as Community Earth System Model (CESM) (Danabasoglu et al., 2020), the Model for Interdisciplinary Research on Climate (MIROC) (MIROC-SPRINTARS) (Takemura et al., 2005, 2009), Weather Research and Forecasting coupled to Chemistry (WRF-Chem) (Grell et al., 2005; Fast et al., 2006), and GEOS-Chem.

Besides, some discussion was added in the revised manuscript, as shown above.

**Comments: I**n line 254, the authors provided an explanation why different structure may lead to different AAOD. However, such explanation is missing in some other analyses. I suggest the authors add similar explanations like line 254 in the descriptions of other results (e.g., why fractal structure produces higher ARI, maybe more solar radiation is reflected by sphere-structure?).

**Response:** Thanks very much for your comments. We have added some explanations like line 254. The reason is that the fractal structure can absorb more light, but the total extinction is not significantly modified. We have added some descriptions in the revised manuscript:

"As explained above, the fractal BC can absorb more light than spherical BC, while the total scattering is not significantly modified. Thus, the fractal BC leads to larger positive clear-sky ARI."

**Comments:** The authors cited several BC forcing values at the beginning of *section 4.4*. Is there any value could be used to compare with the simulation from this study? There are four ARI values for each location in this study, is there any value that is more realistic?

**Response:** Thanks very much for your comments. The ARI in previous studies were also simulation results, and the ARI is difficult to be directly measured. Thus, it is difficult to say which value is more realistic. In this work, we compare the simulation with previously reported values just to show that our simulations generally agree with previous studies, but not to show which value is more realistic. Our main aim is to investigate the sensitivities of ARI to the BC morphology, and which model is more realistic for reproducing the real ARI should be investigated in the future.

Comment: It is confusing to see the "relative variations" of 10.4%-15.3% in the *abstract*. What is the relative variation? Day-to-day variation? Please define it. In *section 4.4*, there is "relative uncertainty", are they the same? In the *conclusions*, it is switched to "relative variations" again.

**Response:** Thanks for pointing it out. In the revised manuscript, we have re-written these sentence in the revised manuscript.

Minor comments:

**Comments:** The writing needs some polishing (e.g., Line 48, contribute to…)

**Response:** Thanks for your comments. We checked the English in the revised manuscript, and all the modifications are marked in the revised manuscript.

**Comments:** Line 95, with a larger D$f$?

**Response:** Thanks for pointing it out. It is indeed "with a larger $D_f$", we have corrected it in the revised manuscript.

**Comments:** Figure 3, lower panel, the four lines are overlapped. You may try to use thicker lines underneath and use thin lines above to make them clearer. It is the same for Figure 6 and 8.

**Response:** Thanks for your comments. We have replotted the figures in the revised manuscript.

---

## Author Comment (AC2)

**Response to the comments of Reviewer #2**

First of all, we would like to thank the two anonymous reviewers for their thoughtful reviews and valuable comments on the manuscript. In the revision, we have accommodated all the suggested changes into consideration and revised the manuscript accordingly. All changes are highlighted in the revised manuscript in **BLUE** in the revision. In this response, the questions and comments of reviewers are in **BLACK** font, and responses are highlighted in **BLUE.** The changes made in the revised manuscript are marked in **RED** font.

General comments:

Materials presented in the manuscript are interesting and well suited to the scope of the current journal. The authors seem to have successfully operated a chain of models to present the simulation results, but their argument needs improvements and additional calculations may be required. Thus, the manuscript should be accepted in the journal, only after the authors revise the manuscript by reflecting the following general and specific comments.

 **Response:** Thanks very much for your comments. The specific responses and revisions are shown in follows.

**Comments:** The title says "regional impacts of ARI". The authors only presented ARI at several grid points, but the regional mean ARI should be presented and discussed. Without the presentation of regional ARI, the current study is not comparing US and China, but comparing a few fire grids in US and a few urban grids in China.

**Response:** Thanks very much for your comments. In the revised manuscript, the regional ARI is presented. Due to huge computations, the regional ARI was calculated using the mean aerosol optical properties, mean cloud properties, and mean albedo, etc.

 We must clarify that the calculation of ARI using mean optical properties would result in some differences from those using temporary optical properties, as the aerosol optical properties would show temporary variations. However, as an estimation for the impacts on the BC morphology, it is still reasonable to make some simplifications as the mean optical properties can also represent the spatial distribution of ARI to some extent. Besides, and similar methods were also used in previous studies (eg. Saleh et al. (2015); Tuccella et al. (2020a)).

**Comments:** There are several types of BC-ARI, but the differences are not clearly stated. For example, the authors mentioned 1.1 W m$^{-2}$ for global mean BC-ARI (Bond

et al., 2013), but actually the value is "total climate forcing". It is totally different from the authors' BC-ARI, that is "clear-sky direct affect". For every ARI value in the manuscript, please be aware which types of ARI you are siting.

**Response:** Thanks very much for pointing it out. We have corrected it in the revised manuscript.

**Comments:** There are clear-sky and all-sky ARI. All-sky ARI is more popular and meaningful. There are also instantaneous ARI and ARI with rapid adjustment. The reviewer understands that the authors' simulation setup cannot derive ARI with rapid adjustment (though they are using WRF-Chem), but all-sky instantaneous ARI can be readily calculated. Impacts of morphology on all-sky ARI should be of interest for readers, too.

**Response:** Thanks very much for your comments. The all-sky ARI was calculated in the revised manuscript. The cloud properties were not simulated in our WRF-Chem simulations. We use the daily-mean cloud optical thickness, cloud effective radius, and cloud cover products from MODIS for the all-sky ARI calculations.

**Comments:** The reviewer does not understand why impacts of morphology on EAE and AAE are important. Presentations of impacts of morphology on regional mean (or regional maximum) clear-sky and all-sky ARI should be much more meaningful.

**Response:** Thanks very much for your comments. The all-sky ARI was calculated in the revised manuscript, and the regional ARI was also calculated using the mean optical properties. The ARI in typical sites are still shown for comparison with previous studies, but the deviations of ARI between non-spherical BC and spherical BC were deleted, as we agree that presentations of impacts of morphology on regional mean    clear-sky and all-sky ARI are much more representative and meaningful.

Specific comments:

**Comments:** Abstract: many important information is missing: "simulation period: season and duration", "clear-sky", "external mixture assumption".

**Response:** Thanks very much for your comments. We have added the information in the revised manuscript, we have re-written the abstract:

"Black carbon (BC) is one of the dominant absorbing aerosol species in the atmosphere. It normally has complex fractal-like structures due to the aggregation process during combustion. A wide range of aerosol-radiation interactions (ARI) of BC has been

reported throughout experimental and modeling studies. One reason for the large discrepancies among multiple studies is the application of the over-simplified spherical morphology for BC in ARI estimates. In current climate models, the Mie theory is commonly used to calculate the optical properties of spherical BC aerosols. Here, we employ a regional chemical transport model coupled with a radiative transfer code that utilizes the non-spherical BC optical simulations to re-evaluate the effects of particles' morphologies on BC shortwave ARI, and the wavelength range of 0.3 - 4.0 $\mu m$ was considered. Anthropogenic activities and wildfires are two major sources of BC emissions. Therefore, we choose the typical polluted area in eastern China which is dominated by anthropogenic emissions, and the fire region in the northwest US which is dominated by fire emissions in this study. A one month-simulation in eastern China and seven-days simulation in the fire region in northwest US was performed. Compared to the spherical BC model, the fractal BC model generally presents a larger clear-sky ARI. Assuming BC particles are externally mixed with other aerosols, the relative differences in the time-averaged clear-sky ARI between the fractal model with a fractal dimension ($D_f$) of 1.8 and the spherical model are 12.1% - 20.6% and 10.5% - 14.9% for typical polluted urban cities in China and fire sites in northwest US, respectively. Furthermore, the regional-mean clear-sky ARI is also significantly affected by the BC morphology, and relative differences of 17.1% and 38.7% between the fractal model with a $D_f$ of 1.8 and the spherical model were observed in eastern China and the fire region in northwest US, respectively. However, the existence of clouds would weaken the BC morphological effects. The time-averaged all-sky ARI relative differences between the fractal model with a $D_f$ of 1.8 and the spherical model are 4.9% - 6.4% and 9.0% - 11.3% in typical urban polluted cities in eastern China and typical fire sites in northwest US, respectively. Besides, for the regional-mean all-sky ARI, the relative differences between the fractal model and the spherical model are less than 7.3% and 16.8% in the polluted urban area in eastern China and the fire region in northwest US, respectively. The results imply that current climate modeling may significantly underestimate the BC ARI uncertainties as the morphological effects on BC ARI are ignored in most climate models. ''

**Comments:** Lns. 53-56, WRF-Chem, FlexAOD, and libRadtran: reference is missing.

**Response:** Thanks very much for your comments. We have added the references in the revised manuscript.

**Comments:** Section 2 and 3 should be combined to one section, "Method".

**Response:** Thanks very much for your comments. We have combined Section 2 and 3 into one section, "Method", in the revised manuscript.

**Comments:** Sect. 2: Which meteorological analysis used for the simulation of China?

**Response:** Thanks very much for your comments. We are very sorry for without clearly clarifying meteorological analysis. For both the simulations in East China and North America, the National Center for Environmental Prediction (NCEP) Global Forecast System's final gridded analysis data set was used to provide the meteorological initial and boundary conditions. The chemical initial and boundary conditions were obtained from the Model for Ozone and Related Tracer, version 4 (MOZART-4). We have clarified it in the revised manuscript.

**Comments:** Lns. 125-128: The reviewer does not fully understand why size distribution of WRF-Chem is not directly used for the optical and radiative transfer calculations.

**Response:** Thanks for your comments. In this revised manuscript, the internally mixing assumption was assumed. Thus, the size distributions of aerosols in WRF-Chem were for the total mixed aerosols, but not BC. In this work, as the first step for using the non-spherical BC in estimating the ARI, we just consider the externally mixed BC, and the internally mixed BC would be considered in the future. Thus, we didn't use the size distributions in WRF-Chem.

**Comments:** Sect. 3: Equations 4-10 are too general and thus you don't need to describe them in the paper. Rather, descriptions or equations describing how to directly calculate the optical properties of fractal agglomerates by MSTM should be elaborated in this section.

**Response:** Thanks for your comments. With the refractive index, wavelength, input position file, which includes the positions and radius of spheres, the MSTM can output the extinction efficiency ($Q_{ext}$), scattering efficiency ($Q_{sca}$), and phase function (P), and the extinction cross-section ($C_{ext}$) and scattering cross-section ($C_{sca}$) were further calculated using Equations 4-5. We have added some descriptions:

"The MSTM can efficiently calculate the optical properties of spheres without intersecting surfaces. The MSTM has high computational efficiency because it theoretically calculates the optical properties of randomly oriented particles without numerically averaging them over different particle orientations. the MSTM can output the extinction efficiency ($Q_{ext}$), scattering efficiency ($Q_{sca}$), and phase function (P) with the refractive index, wavelength, input shape file."

We did not delete Equations 4 -10, as it show how the bulk optical properties of non-spherical BC were calculated.

**Comments:** Ln. 143: What are "the pmom code"? Avoid model-specific terms in a paper.

**Response:** Thanks for your comments. The pmom is a tool available in Libradtran for calculating the Legendre moments. The inputs of pmom are the aerosol phase function and the desired number of Legendre moments. In the revised manuscript, we have added some descriptions of this tool:

"In this work, we used the pmom tool which is available in libRadtran software for calculating the Legendre expansion coefficients. With the inputs of the aerosol bulk phase function and the desired number of Legendre expansion coefficients, the pmom tool can calculate the Legendre expansion coefficients."

**Comments:** Ln. 158: FlexAOD

**Response:** Thanks for your comments. We have corrected it in the revised manuscript.

**Comments:** Ln. 200: What do you mean by "standard atmosphere background"? Instead of using standard atmosphere, the authors should use the atmospheric conditions predicted by WRF.

**Response:** Thanks for your comments. In principle, we should use the atmospheric conditions predicted by WRF. However, this work mainly aims to investigate the effects of BC morphology on ARI, so we use a representative atmospheric profile to eliminate the perturbs of other factors. As the ARI was calculated by the difference between the fluxes with aerosols and without aerosols, the effects of atmospheric conditions should have small impacts on ARI. Thus, we just use the standard atmosphere background. However, after carefully checking the calculations, we found that we have made a mistake in the previous study (we have mistake the aerosol optical properties at the top layer with those at bottom layer). Thus, we have re-conducted the calculations.

**Comments:** Ln. 201: Double periods ".."

**Response:** Thanks for your comments. We have corrected it in the revised manuscript.

**Comments:** Ln. 212, "PM2.5": 2.5 is lowercase here and elsewhere.

**Response:** Thanks for your comments. We have corrected it in the revised manuscript.

**Comments:** Ln. 217, It is a very good idea to compare simulated AOD and AAOD against AERONET in Beijing. Why not other sites in China and US, rather than to compare surface PM$_{2.5}$ only?

**Response:** Thanks for your comments. AERONET sites in our simulation area are rather limited, and other AERONET data for the other site is not available, so we just compare the simulated AOD and AAOD against AERONET in Beijing, and we used the PM$_{2.5}$ comparison for the supplements to show the reasonable predictions. We have clarified it in the revised manuscript.

**Comments:** Ln. 224, "400 ug/m3": it seems the nighttime concentration which does not affect ARI. Please show shortwave and longwave ARI, separately. Longwave ARI could be negligibly small. You may see the phrase "which should have a strong impact on the aerosol radiative effects" is totally wrong. Also, it is just a surface concentration, but the column amount matters for ARI. Reorganize the discussion here.

**Response:** Thanks for your comments. In this work, just shortwave ARI was considered, and the wavelength range was in the range of 0.3 um – 4 um. However, Longwave ARI should be negligibly small. We have clarified it and re-written the sentence in the revised manuscript:

"As shown in Figure 3, the temporal BC concentrations at fire sites can even exceed approximately 400 μg/m$^3$ when the fire occurs, while the BC concentrations are extremely low in other days"

**Comments:** Acknowledgement: please remove FlexAOD here because code availability is in different section.

**Response:** Thanks for your comments. We have removed FlexAOD here.

**Comments:** Figures: please clearly state if the authors use UTC or local time for all time series panels.

**Response:** Thanks for your comments. We have clarified the UTC times in the revised manuscript.

**Comments:** Caption of Fig. 12: probably EAE, not AAE. Probably not lambda=450-850 nm, 850 nm pair but lambda = 450 nm, 850 nm pair.

**Response:** Thanks for pointing it out. We have corrected it in the revised manuscript.

**Comments:** Code availability: code availability should be also stated for WRF-Chem, libRadtran, and MSTM.

**Response:** Thanks for your comments. We have stated the code availability for WRF-Chem, libRadtran, and MSTM.

**Comments:** Data availability: "athour"-> "author". The statement "the data can be requested from the corresponding author" may not be allowed by ACP.

**Response:** Thanks for your comments. We have corrected "author" in the revised manuscript. Besides, we have made the ARI data available in the revised manuscript.

**Comments:** Table S1: please remove (mp_physics), …, (bl_pbl), as those are model-specific terms. Explain acronyms, RRTMG and YSU. Better to include references of each option.

**Response:** Thanks for your comments. We have corrected it and added related references in the revised manuscript.

**Comments:** Table S2: avoid model specific terms. What do a01, a02, a03, and a04 indicate? If it indicates size bins, define the sizes. Same for Table S3. What are those acronyms, for example, orgalk1j?

**Response:** Thanks for your comments. We have revised Table S2 and Table S3 in the revised manuscript.

**Comments:** Fig. S1: boarders (national, province, land/ocean) and symbols are hardly legible. Probably, better to use "white" color for tiny values, instead of "blue".

**Response:** Thanks for your comments. We have replotted the figures, as shown in the following:

[Figure]

Figure 1 the BC concentrations in different regions.